# Tau Seeding Mouse Models with Patient Brain-Derived Aggregates

**DOI:** 10.3390/ijms22116132

**Published:** 2021-06-07

**Authors:** Aiko Robert, Michael Schöll, Thomas Vogels

**Affiliations:** 1Department of Neurodegenerative Disease, UCL Queen Square, Institute of Neurology, University College London, London WC1N 3BG, UK; aiko.robert.20@ucl.ac.uk (A.R.); michael.scholl@neuro.gu.se (M.S.); 2Wallenberg Centre for Molecular and Translational Medicine and the Department of Psychiatry and Neurochemistry, University of Gothenburg, 413 45 Gothenburg, Sweden; 3Department of Psychiatry and Neurochemistry, University of Gothenburg, 413 45 Gothenburg, Sweden; 4Sylics (Synaptologics B.V.), 3721 MA Bilthoven, The Netherlands

**Keywords:** tau, tauopathy, Alzheimer’s disease, animal model, neurodegeneration, seeding

## Abstract

Tauopathies are a heterogeneous class of neurodegenerative diseases characterized by intracellular inclusions of aggregated tau proteins. Tau aggregates in different tauopathies have distinct structural features and can be found in different cell types. Transgenic animal models overexpressing human tau have been used for over two decades in the research of tau pathology. However, these models poorly recapitulate the heterogeneity of tauopathies found in human brains. Recent findings demonstrate that injection of purified tau aggregates from the brains of human tauopathy patients recapitulates both the structural features and cell-type specificity of the tau pathology of the donor tauopathy. These models may therefore have unique translational value in the study of functional consequences of tau pathology, tau-based diagnostics, and tau targeting therapeutics. This review provides an update of the literature relating to seeding-based tauopathy and their potential applications.

## 1. Introduction

Tauopathies are a class of neurodegenerative diseases characterized by intracellular inclusions of aggregated tau proteins. Alzheimer’s disease (AD) is the primary cause of dementia and the most widely investigated tauopathy. AD is additionally characterized by another neuropathological hallmark: extracellular plaques primarily composed of aggregated amyloid-β (Aβ) peptides [1]. Although most AD cases are sporadic, mutations in Aβ-related genes amyloid precursor protein (*APP*), presenilin (*PSEN1*), and *PSEN2* lead to familial AD (fAD). Most so-called “primary” tauopathies, such as Pick’s disease (PiD), progressive supranuclear palsy (PSP), and corticobasal degeneration (CBD), among others, are predominantly characterized by the accumulation of aggregated tau and associated with frontotemporal dementia (FTD) and/or motor syndromes. The different tauopathies differ in their clinical presentations, the neuroanatomical distribution of tau inclusions, the affected cell types, and the occurrence of structurally distinct tau fibrils that are composed of different tau isoforms [2].

Importantly, the accumulation of hyperphosphorylated tau aggregates has been found to directly correlate with neurodegeneration and cognitive decline, in AD as well as in other tauopathies. Neuropathological and in vivo positron emission tomography (PET) findings have described the spatiotemporal progression of tau pathology to mirror that of the clinical symptoms observed in tauopathy patients [3,4,5,6]. Additionally, tau pathology along Braak stages in AD appears to progress along neuroanatomically connected brain regions [7,8,9]. This has been suggested to occur by means of spreading of tau aggregates to synaptically connected neurons and subsequent seeded aggregation of physiological tau proteins in previously unaffected cells [10,11,12,13]. This hypothesis therefore provides a compelling explanation for the observed propagation of tau pathology in AD patients [2].

Several aspects of tau pathology in tauopathy patients have been successfully modelled in transgenic mice overexpressing human tau, including tau-associated processes such as neuroinflammation, synapse loss, neurodegeneration, and cognitive impairment [14,15]. Moreover, these models have enabled the recapitulation of specific pathogenic processes, thereby significantly contributing to our understanding of human tauopathies [16]. However, they may provide limited translational value in the research of sporadic tauopathies, as most overexpress tau bearing familial FTD-related mutations. Additionally, *post-mortem* human tauopathy brains show a high degree of heterogeneity of tau pathology (reviewed in [17]), which is poorly recapitulated in transgenic animals—both in terms of the structure of tau aggregates and the affected cell types.

In recent years, efforts to develop animal models with improved translational value have led to the emergence of seeding-based animal models injected with tau aggregates derived from patients with different tauopathies [18]. These models represent a significant advance in several aspects. Firstly, they recapitulate the structural features of the injected tau aggregates [19]. Secondly, tau aggregates can be found in the same cell types as the donor tauopathy [20,21]. Finally, patient-derived seeding models could potentially be more translationally relevant models in that they are not restricted to a subset of tauopathies. Instead, brain extracts from patients affected by any tauopathy can be injected. These models allow for the induction of a localized tau pathology via injection of brain extracts in targeted brain regions, thus enabling investigations of tau spreading. This also allows for therapeutic testing of tau-targeting drugs specifically directed against patient-like tau aggregates. Pre-clinical models with better translational value are important, given the pressing urge for a disease-modifying treatment for AD and other tauopathies, especially in the face of recent failures of Aβ and tau-targeting drugs in expensive and time-consuming later stage clinical trials [22].

This review provides an overview of the recent literature on patient-derived tau seeding-based animal models and describes their significance in improving our understanding of tau pathology and testing tau-based therapeutics.

## 2. Tau Protein Properties in Health and Disease

### 2.1. Tau Physiology

Tau is a soluble protein encoded by the microtubule-associated protein tau (*MAPT*) gene, located on chromosome 17q21.31. It can mainly be found in neurons and plays a role in promoting microtubule assembly and stability, among other functions. Structurally, tau proteins can be divided into several sub-domains: the N-terminal domain, the proline-rich mid-domain, the microtubule-binding region (MTBR), and the distal C-terminal domain. In the human central nervous system, *MAPT* mRNA is subject to alternative splicing at several locations (exons 2, 3, and 10), the product of which results in six different tau protein isoforms. These differ in the number of N-terminal inserts, ranging from zero to two, and the presence of three or four microtubule-binding repeats (R) (Figure 1). As such, tauopathies are usually termed 3R, 4R, or 3R/4R, according to the number of MTBRs incorporated in their characteristic tau inclusions [23,24,25,26,27].

Mutations in the *MAPT* gene are a known cause of certain familial forms of FTD. Over 50 mutations have been associated with an increased risk of tau pathology and the onset of clinical symptoms. Most mutations are located in exons encoding the MTBR, adjacent exons, or the intronic sequences surrounding them. Moreover, *MAPT* mutations have been shown to alter the affinity of tau for microtubules, to affect splicing to favor 4R isoforms or to promote aggregation-prone conformations of tau [28,29,30]. There are also different *MAPT* haplotypes, of which H1 has been associated with an elevated risk of developing a number of primary tauopathies relative to those bearing the H2 haplotype. This seems to occur mainly in Caucasian patients and could be attributed to a slight overexpression of tau [31,32,33,34,35].

Additionally, tau proteins are subject to multiple post-translational modifications (PTMs), including phosphorylation and truncation. Elucidating the functional implications of PTMs remains an active topic of investigation both in terms of tau biology and pathophysiology [36,37,38]. A number of studies have pointed at the critical role played by some PTMs in pathogenesis, for instance by influencing tau proteins’ proneness to aggregation. Phosphorylation remains the most widely investigated PTM to date. Given that tau phosphorylation status greatly influences physiological protein function, the hyperphosphorylated tau proteins characteristic of tauopathies could be key to driving aggregation, via a yet unknown mechanism [39,40]. Gaining a deeper understanding of site-specific tau phosphorylation in the healthy brain may help elucidate its pathogenic roles.

Other PTMs have also been tied to tau aggregation mechanisms. For instance, acetylation by the SIRT1 enzyme is thought to be protective against tau aggregation in vitro, whilst deacetylation by HDAC6 enhances tau phosphorylation and accumulation. Moreover, a reduction in *SIRT1* expression but a heightened *HDAC6* expression have been observed in AD patients [41,42,43]. Truncation at several sites in the C-terminal region of the tau proteins, or within the proline-rich domain, have also been associated with increased tau accumulation and neurotoxicity in rodents [44,45,46,47,48].

Taken together, several lines of research present PTMs as key factors contributing to pathogenesis in AD and other tauopathies. Of note, most studies have identified PTMs and elucidated their functional consequences in vitro, or in cell or animal models relying on tau overexpression. Therefore, the use of more translationally relevant models to determine the exact mechanisms underlying PTMs and their implications on a functional level may be required.

### 2.2. Tau Pathophysiology

Cellular deposition of insoluble tau aggregates is a hallmark of neurodegenerative diseases termed tauopathies. Under pathological conditions, tau monomers can assemble into toxic oligomers. These can in turn grow into insoluble fibrils by means of β-sheet formation in the MTBRs of misfolded tau proteins. Subsequent events include their continued aggregation and elongation, due to the seeding of physiological tau proteins, resulting in the formation of soluble oligomers and in turn, insoluble fibrils. Together, these mechanisms are referred to as templated misfolding or seeded nucleation [49,50]. In AD, insoluble fibrils can present in the form of paired helical filaments or straight filaments [2]. Tau filaments in AD can further accumulate within neurons to form neurofibrillary tangles (NFTs) in neuronal cell bodies and neurofibrillary threads in axons and dendrites. In contrast, other primary tauopathies present with structurally distinct tau fibrils, and most of these diseases exhibit glial tau pathology in addition to neuronal pathology [2].

It is still an outstanding question which forms of tau are the predominant seed-competent species in tauopathy patients. Misfolded monomers, soluble hyperphosphorylated tau aggregates, and short fibrils have all been proposed to be responsible for tau seeding [29,51,52]. Interestingly, differences in seed-competence of soluble hyperphosphorylated tau species were correlated to clinical heterogeneity in AD patients [51,52]

In 1991, Braak and Braak defined stages describing the progression of tau pathology in AD across neuroanatomical regions [53]. Braak stages I and II indicate the presence of pathology in the entorhinal cortex (EC) or rostral medial temporal lobe. Braak stages III and IV indicate pathology progression to limbic regions. The most advanced Braak stages V and VI describe tau pathology across the neocortex, with the primary motor and sensory cortical areas less affected. To this day, Braak staging remains widely used to describe the progression of tau pathology, following suggestions from PET studies that its distribution closely correlates with disease symptoms and progression [4,5,6]. However, recent tau-PET studies have nuanced these widely accepted Braak stages by revealing heterogeneous tau deposition patterns among patients and suggesting that they corresponded to distinct clinical profiles [7,54,55,56,57]. Franzmeier and colleagues [58] recently developed a model to assess such tau spreading based on connectivity to predict disease progression whilst accounting for inter-individual variability in spreading patterns.

Moreover, growing evidence suggests that this spread occurs via transmission of pathological tau from a donor cell to a recipient cell, such that recruitment of physiological tau in the latter generates new deleterious seeds. This pathological spread provides a good explanation for the stereotypical progression pattern of tau pathology observed in tauopathies [2,10,11,13,59].

Tau secretion from synaptic terminals can occur via one of several non-exclusive pathways. In addition to release resulting from normal synaptic activity [60,61,62], tau secretion also occurs via several less conventional molecular mechanisms. For example, tunneling nanotubes could transfer tau aggregates between cells [63]. Recently, accounts of tau secretion across the plasma membrane in the absence of vesicles have been reported [64,65]. Pore formation in the plasma membrane has also been suggested as another route through which tau can enter the extracellular space [66].

Once in the extracellular space, pathological tau proteins need to be internalized into a neighboring recipient cell. This can also occur in several ways, including dynamin-dependent bulk endocytosis [67], macropinocytosis following heparan sulphate proteoglycan binding on the plasma membrane [68,69], and/or through the interaction with several specific receptors located at the cell surface, such as low-density lipoprotein receptor-related protein 1 (LRP1) [70,71,72,73]. Tau aggregates then leak into the cytosol by rupturing the endosomal membrane, where they can interact with physiological tau [74,75]. The relative contribution of each of these secretion and uptake mechanisms to disease remains largely unknown. It is possible that this is also different for distinct tau species, which could contribute to their unique spreading patterns and cell-type specificity.

Finally, pathological tau proteins in different tauopathies have often been found to exhibit abnormal PTM patterns. For instance, in AD patients, phosphorylation was suggested to predispose deleterious species to the aggregation process, with a gradual increase in other PTMs following the progression and maturation of tau pathology [76]. Tau pathology therefore does not only progress by propagating to other brain regions but also progresses over time within affected cells [77].

### 2.3. Different Tauopathies Exhibit Different Pathological Features

Interestingly, the abnormal tau proteins forming the signature intracellular inclusions of tauopathies have been found to be structurally distinct, and to exhibit differential cell-type specificity. These diseases can be broadly subdivided into three main groups according to the number of microtubule-binding repeats in the tau proteins constituting the inclusions: 3R, 4R, or mixed 3R/4R [26].

AD and chronic traumatic encephalopathy (CTE) are tauopathies comprising both 3R and 4R tau fibrils, which form NFTs, as well as some degree of astrocytic tau pathology in the latter. Conversely, PiD is an example of a tauopathy presenting with 3R tau aggregates in the form of neuronal inclusions, termed Pick’s bodies. Several tauopathies are characterized by 4R tau fibrils, including CBD and PSP, which are characterized by neuronal as well as oligodendroglial and astroglial inclusions. Globular glial tauopathy (GGT) and argyrophilic grain disease (AGD) also both present with 4R tau aggregates, in neurons and oligodendroglia in GGT, and in the form of argyrophilic grains and astroglial, oligodendroglial and neuronal deposits, in AGD (reviewed in [17]).

The recent emergence of cryogenic electron microscopy (cryo-EM) has enabled the characterization of the structural differences in tau aggregates in the different tauopathies. This tool allows for the visualization of the core of tau filaments derived from tauopathy patients’ brains at the atomic level. This provides unprecedented evidence that each tauopathy is characterized by a unique tau filament fold. The core of PiD tau filaments was revealed to consist of nine β-strands spanning MTBRs R1 to R4, and nine additional amino acids after R4 [78]. CBD is the first 4R tauopathy whose tau filament core has been characterized, comprising 11 β-strands forming a four-layered structure which includes the last R1 residue, the whole of R2 to R4, and 12 amino acids after R4 [79]. Somewhat similar to AD, the CTE tau core consists of eight β-strands comprised of 3R- and 4R-tau residues forming two C-shaped protofilaments [80]. Interestingly, primary age-related tauopathy (PART) has been identified as presenting an identical tau filament core to AD cases, suggesting that it may be an AD variant [81]. Additionally, these studies have further suggested that disease-specific fibrils have the same core structure among patients with the same tauopathy [78,82,83] (Figure 1).

Taken together, several lines of research point at the idea that tauopathies differ in their pathological mechanisms due to several factors: the involvement of different tau isoforms, the generation of different ultrastructural aggregates, and their formation in distinct cell types. These differences could account for the phenotypic diversity among tauopathies, and therefore underlie their heterogeneity.

## 3. Animal Models of Tau Propagation

Considerable focus has been placed into in vivo modelling of tauopathies using rodent models. The strength of these models compared to cellular models is that rodents have a brain that is in many ways similar to humans. Early efforts were limited to transgenic (Tg) mouse models exhibiting widespread tau pathology in neurons, which are still widely used given their robust tau pathology, neuroinflammation, neurodegeneration, and behavioral impairments (reviewed in [84]). However, given the recent focus on the mechanisms of tau propagation, there has been an increasing interest in animal models with localized tau pathology. These models can be roughly classified into four categories: localized Tg tau models with expression of mutated tau restricted to the EC, adeno-associated virus (AAV)-based models with inducible tau expression, seeding-based models injected with recombinant tau fibrils, and seeding-based models injected with human brain-derived tau fibrils. This section will provide a brief description of all aforementioned models, with a specific focus on patient brain-derived tau seeding models in the next section.

### 3.1. Transgenic Mouse Models

Three broad types of Tg mouse models exist: those expressing 3R isoforms, those solely expressing 4R isoforms, and models expressing all six wild-type human isoforms in the absence of murine tau [19,84,85]. Tg models expressing one mutated tau isoform, most often 4R, have been widely used because they exhibit rapid and robust tau pathology. They often carry a transgene containing the P301L or P301S *MAPT* mutation, which are known familial FTD-associated mutations [86]. Other mutant forms such as A152T or K280del have also been used [87,88]. Additionally, various truncation sites on the tau protein have been exploited to promote aggregation in mice [46,48,89]. At the microscopic level, Tg mice develop neuronal inclusions resembling NFTs in several brain regions, in addition to various degrees of neuroinflammation and neuronal loss [15,90].

Although most Tg mouse models have focused on tau pathology in neurons, several pioneering studies have also selectively modelled astrocytic or oligodendrocytic tau pathology [91,92,93]. However, major limitations of these studies are that the glial promotors used led to relatively weak human tau expression, and the resulting tau pathology preferentially affected the spinal cord. This resulted in late-onset tau pathology and functional deficits that poorly recapitulated primary tauopathies [91,92,93]. Therefore, this approach could be used in future studies, upon identification and characterization of more practical promoters. Importantly, transgenic models are associated with phenotypic artefacts not related to tau pathology, caused by the disruption of endogenous genes and genetic drift [94]. The genome and phenotype of both existing and new transgenic models therefore need to be carefully characterized. Furthermore, sex differences have been observed in transgenic tauopathy mice [95]. However, these are often not explicitly reported or investigated, which could be an important focus of future research.

### 3.2. Transgenic Mouse Models with Localized Expression

Tg models with localized tau pathology were designed to selectively express mutated human tau in the EC, which along with the locus coeruleus, is one of the earliest brain regions affected by tau pathology in AD—corresponding to Braak stages I and II [12,13,96]. They are obtained by crossing a line expressing the tetracycline transactivator (tTA) under the neuropsin promoter, with a responder line, which only expresses the FTD-linked P301L mutation in the presence of tTA. Deposited tau proteins in these mice exhibit the pathological features one would expect, including hyperphosphorylation, abnormal tau folding, and accumulation of predominantly neuronal aggregates [13]. However, despite the successful induction of tau pathology affecting the EC and its connections, which are implicated in several types of memory, these mice do not display measurable cognitive deficits [96].

Nonetheless, Tg models with selective transgene expression in EC have successfully recapitulated the observations made by Braak and colleagues that tau spreads in a progressive and stereotypical pattern across neuroanatomically connected regions, although the possibility that such spread results from mutated human tau seeding mouse tau in cells overexpressing tau cannot be excluded. In addition, the progression of tau pathology appeared to develop in an age-dependent manner, although this could also be the result of prolonged transgene overexpression [12,13,96].

Whilst these models have proven invaluable in the in vivo modelling of tau propagation, especially in the context of AD—whereby tau pathology initiates in the EC—they are imperfect in several aspects. First and foremost, these models recapitulate early events in AD-related tau propagation from an anatomical perspective, but the structure and conformation of the tau aggregates is likely to be distinct from those found in the brain of AD patients. They have limited translational value as they solely express one tau isoform (usually 0N4R), whereas tauopathies like AD contain aggregates incorporating both 3R and 4R isoforms. In addition, they rely on the artificial overexpression of tau with human familial FTD-linked mutations [12,13,96]. The different tau conformations observed in human tauopathy patients are likely to play a key role in the differential release, uptake, and therefore spreading route taken by tau proteins.

### 3.3. Viral Expression of Human Tau in Wild-Type Mouse Models

More recently, AAV-based tauopathy induction has been developed as a means of inducing tau pathology in wild-type (WT) mice. These models present as an attractive alternative to the Tg models in that they entail several key improvements. Firstly, they induce rapid spread of tau pathology following injection in the EC or hippocampus [97,98]. Secondly, AAV vectors can be injected into any brain region. This minimizes the induction of tau pathology in the brainstem and spinal cord like many transgenic animals, which is associated with undesirable functional consequences (e.g., paralysis). Thirdly, they can include a cell type-specific promoter to select the cell type bearing the tau pathology (e.g., astrocytes) [99,100]. Likewise, they can include a fluorophore (e.g., green fluorescent protein (GFP), which labels transduced cells, thereby differentiating them from recipient cells. Fourthly, they have the considerable advantage of being able to induce tau pathology in a range of genetic backgrounds, thus circumventing the costly and slow process of cross-breeding [101]. Finally, they enable the investigation of spreading patterns in models bearing pro- and anti-aggregant tau mutations [102]. Typically, AAV-based models comprise the human FTD-linked mutant P301L tau. Local expression of the AAV in the EC has been shown to induce propagation of human tau, which could take place even in the absence of endogenous mouse tau [98]. Other studies have also used AAV-based models to study the role of intracellular pathways in the spread of tau pathology [103].

Interestingly, AAV-based mouse models have provided direct accounts for the age-dependency of tau pathology progression, which is not observable in most Tg models, as the majority express the transgene from birth. Wegmann et al. (2019) [104] virally expressed human P301L mutant tau into the EC of both young and old mice, demonstrating exacerbated tau spreading and misfolding in aged animals—with a tau spreading rate nearly twice that of younger mice. These findings point at the likelihood that age represents a favorable environment for tau spread [104].

Despite the considerable advantages listed above that AAV-based models hold, they remain imperfect in their translational value. The resulting fibrils differ from those found in sporadic tauopathies, because they rely on the use of familial FTD-associated mutations. However, this was overcome by viral expression of non-mutated truncated human tau protein in the hippocampus of WT mice. This resulted in hyperphosphorylation of tau, accumulation of NFTs that also included mouse tau, and spread of human tau across the hippocampal circuit [105]. Whether this method induces tau filaments with a structure similar to AD remains to be determined.

### 3.4. Seeding with Tg Mouse Brain Lysates

In 2009, Clavaguera and colleagues [10] conducted a key experiment regarding tau seeding and spreading, which eventually paved the way for studies involving intracerebral injection of patient-derived tau into mouse models. They injected tau filament-containing brain homogenates from a Tg mouse model that expresses the 0N4R human tau isoform with the FTD-linked P301S mutation into the hippocampus and overlying cerebral cortex of Tg mice overexpressing a single WT human tau isoform (2N4R). This seminal study provided the first evidence that injected tau fibrils induce the formation of pathological tau aggregates, which progressively spread to brain regions anatomically connected to the injection sites. Tau pathology was observed in different cell types, with aggregates present in the form of NFTs, neuropil threads, and coiled bodies in oligodendroglia. Moreover, further analysis suggested that these aggregates were composed of insoluble, phosphorylated tau [10].

Similarly, Ahmed et al. (2014) [106] proceeded to inject brain extracts of 5.5-month-old P301S transgenic mice for seeding into the brain of mice from the same line at 2 months of age. This resulted in a considerably more rapid and robust tau pathology induction than that obtained in the previous study, with the formation of neuronal inclusions in the form of neuropil threads and NFTs starting 2 weeks post-injection in the ipsilateral side, and after 1 month on the contralateral side (versus 15 and 18 months reported in [10]). Importantly, these series of experiments confirmed the hypothesis that pathological tau spread occurred via synaptic connectivity rather than proximity. This idea was further supported by the formation of tau pathology in the white matter tracts connecting regions affected by tau pathology [106].

Despite their critical importance, these models present with a major limitation due to the presence of a single tau isoform (4R in this case). Moreover, their relevance in recapitulating human disease is questionable due to the transgenic nature of the injected mouse brain homogenates. Although tau immunodepletion was performed, this method could not rule out the possibility that the tau pathology observed in these mice solely resulted from the injected material [10].

### 3.5. Seeding with Recombinant Tau Fibrils

Another means to induce tau pathology in mice is via intracerebral inoculation of synthetic so-called preformed fibrils (PFFs). These are generated by assembling purified recombinant full-length (i.e., 2N4R) or truncated 4R-tau, which fibrillize in the presence of heparin. The first in vivo account was provided by Iba et al. (2013) [107], whereby endogenous murine tau from mice of the PS19 line was incorporated into pathological neuronal tau aggregates. This was followed by tau spread across brain regions neuroanatomically connected to the injection site. This study was the first of many to demonstrate that tau fibrils alone were capable of inducing tau pathology in young Tg tauopathy mice [107]. Further studies utilizing this system also demonstrated induction of hyperphosphorylation and aggregation of tau in a variety of transgenic animals [108,109]. Furthermore, one study observed the selective loss of neurons in the CA1 region of the hippocampus following intracerebral injection of PFFs in the brain of P301L Tg mice [108].

However, Guo et al. (2016) [110] failed to induce tau pathology in WT mice upon injection of PFFs. Conversely, inoculation with human AD brain extracts was sufficient to produce robust pathology, revealing differential conformational features between synthetic fibrils and AD brain-derived seeds. On a structural level, recombinant tau fibrils differ dramatically from patient-derived fibrils—highlighting their highly limited translational value [111]. Therefore, synthetic tau fibrils have largely fallen out of favor compared to patient-derived tau seeding models, which will be the subject of the below section.

## 4. Patient-Derived Tau Seeding Models

Recent years have seen the emergence of studies modelling tauopathies in vivo by injection of human patient-derived tau into mice (Table 1). These models have greater translational potential compared to Tg tau mice, viral models, or seeding by injection of recombinant tau fibrils. Patient-derived tau seeding models circumvent the need for artificial overexpression of mutated or truncated versions of human tau. They have been widely used to induce AD-like tau pathology, but also to mimic the histopathology of other tauopathies [2]. The main strength of these seeding models is their ability to recapitulate the distinct disease-specific tau fibrils, as well as recapitulating the cell-type specificity of tau inclusions found in the different tauopathies [18]. Patient-derived homogenates have the considerable advantage over recombinant fibrils that they comprise the human tauopathy core structure and multiple human tau isoforms, in addition to bearing any complex hyperphosphorylation or other PTM patterns found in human tau fibrils [76]. Moreover, they can be injected into a range of rodent models, including tau Tg mice, WT mice, or mice with a humanized *MAPT* sequence [21,110,112]. In addition, they can also be injected in other non-tau Tg models, for example, those that recapitulate other aspects of the disease (e.g., amyloid pathology) or express tauopathy-associated risk gene variants (e.g., *TREM2*) [113,114]. Importantly, AD-like tau pathology has been successfully induced in Tg models of Aβ pathology, which can be considered a strong translational mouse model of AD [113,115].

### 4.1. Inoculation of Tg Mice Overexpressing Human Tau with FTD Mutation

Since the first demonstration of brain-derived tau seeding models [10], there have been several studies reporting the rapid and robust induction of pathological tau seeding and spreading in Tg mice bearing a human FTD-associated *MAPT* mutation following inoculation with patient-derived tau from individuals presenting a range of tauopathies. Brain extracts from AD or CBD patients induced tau pathology in as little as 4 weeks after intracerebral injection in the PS19 line, which expresses human tau with the P301S mutation [21]. Over time, pathological tau deposition increased in recipient animals and progressively spread to brain regions anatomically connected to the injection site, both ipsi- and contralaterally. Tau pathology was found to be highly specific to the human pathology being induced, such that immunohistochemical findings revealed the presence of disease-specific inclusions using both anti-3R- and 4R-tau monoclonal antibodies directed against AD- and CBD-tau pathologies, respectively. Specifically, AD-tau injected mice had NFT-like inclusions in neurons, whilst CBD-tau inoculated mouse brains also exhibited astrocytic and oligodendroglial tau inclusions. The amount of tau pathology was dose-dependent, with higher concentrations of injected pathological tau inducing a heavier pathological tau burden. Not only did tau load increase with higher doses of CBD- and AD-tau, its distribution across connected brain regions—ipsi- and contralaterally—also increased [21]. Taken together, these findings reinforced the idea that the distinct tau fibril conformations encountered in different tauopathies could be responsible for their heterogeneous pathological and clinical presentations.

Recently, a similar method was employed by intracerebrally injecting Tg P301S mice with cerebrospinal fluid (CSF) instead of brain homogenates from patients presenting with probable AD or mild cognitive impairment due to AD [116]. CSF inoculation produced similar effects to those described above, suggesting that CSF also contains biologically-active tau seeds. Thus, this model could be used in future studies to elucidate the different seed-competent tau species present in CSF, and study potential differences between CSF and brain-derived tau. This model could also be used to investigate the disease mechanisms resulting from the potential spread of tauopathy-specific seeds or at different Braak stages.

Interestingly, Gibbons et al. (2017) [109] developed a mouse line carrying the FTD-linked P301L *MAPT* mutation paired with GFP to sensitively visualize tau aggregation in vivo. This novel technology could be employed to elucidate some of the mechanisms underlying tau aggregation and pathology spreading more sensitively and less labor intensively than histological staining of brain slices. Importantly, the earliest stages of tau aggregation could potentially be visualized directly in cells using in vivo two-photon imaging. Similarly, it could be employed when culturing primary neurons as well, without the need for additional transfection or transduction of other tau aggregation biosensors [109].

### 4.2. Injection in Tg Mice Overexpressing A Single WT Human Tau Isoform

One of the early accounts of pathological tau induction with brain extracts derived from tauopathy patients was the study of Clavaguera et al. (2013) [20]. They injected the same ALZ17 mice as in their original study—overexpressing a single WT human 4R tau isoform—with brain homogenates from AD, PART, PiD, AGD, PSP, and CBD patients. As a result, the hallmark lesions of each tauopathy were successfully recapitulated, in accordance with their morphological diversity and cell-type specificity, with progressive spread in all cases [20]. However, they failed to replicate the hallmark tau inclusions from the 3R tauopathy PiD. Since the ALZ17 line solely expresses 4R tau, the injected fibrils lack the required availability of human 3R tau proteins to form Pick’s bodies. Additionally, as AD comprises both 3R and 4R tau, its resulting fibrils can only consist of 4R tau isoforms in this model. Therefore, the resulting AD tau-induced pathology in this line does not fully recapitulate the diversity of tau fibrils in AD. Consequently, the authors suggested that different tau strains could be at the basis of the heterogeneity of tauopathies. Further, they presented this model as a potential experimental system to inject brain extracts from patients carrying distinct sporadic tauopathies to elucidate tau seeding and spreading mechanisms.

Pathological tau seeding has been associated with cognitive decline in a recent study reporting that mice overexpressing WT human 2N4R tau (Tg601) inoculated with AD brain homogenates exhibited inclusions containing hyperphosphorylated tau and performed poorly in the Barnes maze test 17 to 19 months post-injection [117]. This impairment in memory and learning was further correlated with an increased microglial cell count in both hippocampi of these mice, in the absence of overt neurodegeneration [117]. Future studies may help elucidate the potential functional and cognitive consequences of tau seeding in similar models.

### 4.3. Injection into WT Mouse Models

WT mice have the considerable advantage over Tg mouse models that seeds from human brains can be injected without the need for breeding the tau transgene along with other transgenes of interest. This is particularly useful when investigating tau pathology in combination with another disease aspect, such as amyloid pathology in the case of AD. This model also does not rely on the artificial overexpression of tau, thereby preventing potential overexpression artefacts [118].

In the first successful study, Guo et al. (2016) [110] administered AD brain-derived tau fibrils intracerebrally in WT mice. As a result, tau proteins accumulated to ultimately form inclusions across those brain regions anatomically connected to the injection site. Nonetheless, the extent of pathological tau distribution obtained in these mice was more restricted than typically observed after injections in Tg animals [110]. Tau pathology upon patient tau injection in WT animals also developed relatively slowly and, interestingly, was decreased at 9 months after injection [110]. This latter finding could be potentially explained by intracellular tau fibril degradation, as described in a recent study using an organotypic brain slice model [119].

In the following years, a host of tauopathies were induced in WTs, from 3R and 4R tauopathies AD and PART, through to 4R tauopathies including aging-related tau astrogliopathy (ARTAG), GGT, PSP, CBD, AGD, and familial FTD linked to the P301L *MAPT* mutation [18,110,120,121,122,123,124,125,126,127].

Narasimhan et al. (2017) [122] observed signs of glial involvement in pathological tau transmission following the injection of PSP and CBD patients’ brain extracts. The cell-type specificity of tau aggregate formation in each tauopathy was maintained, such that tau pathology was present in astrocytes in PSP-injected mice, for example. Interestingly, the pathological tau distribution pattern differed according to the initial injection site, suggesting that the brain region at which seeding is initiated, along with its neuroanatomical connections, determine the regional distribution of distinct tau strains, rather than the structural features of the seeds themselves.

More recently, Ferrer et al. [123,124,125,126,127] conducted a series of experiments consisting of the unilateral inoculation of sarkosyl-insoluble fractions from a range of tauopathies into the hippocampus of WT mice [124,125,126]. Ferrer and colleagues found some degree of morphological and biochemical fluctuations across GGT patients [123]. Despite the successful seeding and spreading of pathological tau following injection of brain extracts derived from human patients, they reported a failure to recapitulate some of the hallmark features of tauopathies, such as globular oligodendroglial and astroglial inclusions in GGT or thorn-shaped astrocytes upon ARTAG-tau inoculation. Instead, neuronal and oligodendrocytic inclusions were observed [123,125].

All in all, the injection of patient tau homogenates into WT mice has its practical benefits. However, one important caveat is the considerable difference between human and mouse tau proteins [128]. First, adult mice brain mainly comprise 4R tau isoforms, whilst adult humans express equal levels of 3R and 4R in the brain. Of note, the presence of 3R tau isoforms has been suggested as a key element in the maturation of tau pathology [129]. In addition, the N-terminus of mouse tau proteins largely differs from the N-terminus in human tau (reviewed in [26]). Mouse tau may therefore not be entirely compatible with the injected human tau seeds [112].

### 4.4. Inoculation into Mice with a Humanized MAPT Sequence

The emergence of human-like tauopathy mouse models bearing all six non-mutated human tau isoforms in the absence of mouse tau has enabled for the first time the recapitulation of all tauopathies in a single Tg model. Humanization of mouse tau overcomes the incompatibility of 4R tau Tg models with PiD fibrils, for example, with greater compatibility of human tau aggregates with human tau than with previously-mentioned mouse tau. To this day, they are considered the best translational models for sporadic tauopathies in that the resulting pathology consists entirely of human tau isoforms.

Humanized tau mice have therefore been used to investigate the role of tau hyperphosphorylation. hTau mice were generated by first knocking out mouse tau sequences and then cloning in the entire non-mutant human *MAPT* sequence, as an early humanization attempt [85]. Hu et al. (2016) [130] injected hTau mice with phosphorylated tau (P-tau) derived from AD patients’ brains. As expected, this triggered the formation of NFTs and neuropil threads in and around the injected site. Interestingly, this only occurred upon injection of P-tau. Dephosphorylation following protein phosphatase 2A (PP2A) treatment resulted in a strong decrease in pathological tau load, as well as morphological changes resembling argyrophilic grains. These findings led to the conclusion that abnormal phosphorylation of tau fibrils was necessary for maintaining seed-competence [130]. Several studies have since identified tau hyperphosphorylation at specific residues on the protein following inoculation with AD- and CBD-derived material into hTau mice [131,132].

He et al. (2020) [19] developed a Tg human-like tau model expressing equal ratios of 3R and 4R tau isoforms, as is the case in the adult human brain, termed 6hTau. This line represents an improvement compared to the original hTau model, which tended to express elevated 3R tau levels compared to 4R [85]. The 6hTau mice were inoculated with AD-, PiD-, CBD-, and PSP-tau extracts, which induced the spread of tau seeds bearing strain-specific structural and seeding properties, such that AD-tau injection induced a mixed 3R and 4R tau pathology, PiD-tau selectively induced 3R tau pathology, and CBD- and PSP-tau injection selectively induced 4R tau pathologies—as determined by staining with 3R and 4R-specific antibodies. In addition, the cell-type specificity of each tauopathy was maintained. Importantly, injection of brain lysates from these seeded 6hTau mice into naive 6hTau mice maintained these strain specificities [19]. Moreover, in vitro amplification of patient-derived tau seeds from AD, CBD, and PSP with 2N4R recombinant tau was performed, which faithfully retained strain-dependent pathogenic characteristics when injected into 6hTau mice. Interestingly, the amplified AD fibrils could recruit both 3R and 4R tau, even when they consisted only of 4R tau isoforms [133].

These models have therefore been widely used to elucidate the distinct neuropathological properties differentiating mostly sporadic, but also some familial tauopathies [19,132,134]. Moreover, they present as a model with great translational potential for testing tau-targeted drug treatments (see Section 5.6) [135]. Importantly, by comparing the same AD-tau injection in WT mice and mice with a humanized tau sequence, human tau was shown to be a better substrate for human-derived tau seeds than mouse tau [112]. Thus, humanized tau mice are a strong translational model in terms of recapitulating primary tauopathies, where there is no requirement for the concomitant modelling of Aβ pathology.

### 4.5. Towards More Integrative AD Models

AD is characterized by two main neuropathological hallmarks: Aβ and tau pathologies. Thus far, although both pathologies have been successfully modelled separately in vivo, attempts to integrate both aspects into a single animal model have been imperfect. For instance, 3xTg mice bearing three familial dementia mutations have been the first mouse models to bear both Aβ and tau pathologies [136,137]. These mice have two familial AD mutations in the Aβ-related *APP* and *PSEN1* genes and one P301L mutation in 4R *MAPT*, which is only found in familial FTD. Therefore, despite the presence of Aβ plaques, this model is unlikely to bear tau pathology mimicking that found in AD patients.

The mounting use of AD patient-derived brain extracts has enabled the induction of AD-like tau pathology in Aβ mouse models. This provides a means to re-assess the relationship between these two factors, whilst recapitulating all three forms of pathological tau found in AD patients: NFTs, neuropil threads, and neuritic plaques or dystrophic neurites around Aβ plaques. He et al. (2018) [113] suggested that Aβ plaque formation was necessary but not sufficient for tau spread, upon AD-tau injection in two Aβ mouse models (5xFAD and *APP* knock-in mice). Several lines of evidence support the view that the presence of Aβ plaques creates an environment which favors seeding and spreading of endogenous tau [115]. Moreover, there have been suggestions that Aβ plaque burden positively correlates with dystrophic neurites, but not NFTs, following AD-tau injection [113]. Inoculation of human AD tau seeds into the brain of an Aβ mouse model further suggested a positive relationship between the severity of tau lesions and lighter microglial load with synapse loss and cognitive impairments, in accordance with human PET findings [138].

Saito et al. (2019) [112] developed a double knock-in (KI) mouse model by cross-breeding *MAPT* KI mice, bearing humanized tau, with single *APP* KI mice. These mice are characterized by Aβ pathology and express all six isoforms of human tau, with 3R tau isoforms slightly more abundant compared to 4R isoforms. In line with the previous findings, greater tau phosphorylation and accumulation into dystrophic neurites were observed in the presence of Aβ pathology (i.e., double KI mice) than in single KI. Injected AD brain extracts into this mouse model showed stronger interaction with humanized tau compared to mice with endogenous mouse tau, presenting this model as an attractive candidate for elucidating AD disease mechanisms by injection of patient-derived brain homogenates. Importantly, targeted replacement of the murine *Mapt* sequence with a humanized version eliminates any potential overexpression artefacts, as human tau is expressed at endogenous levels [112].

Interestingly, intravenous injection of AD-tau into 5xFAD mice resulted in signs of neuroinflammation in the form of astro- and microgliosis, increased Aβ burden, and elevated levels of hyperphosphorylated tau in plaque-associated dystrophic neurites [139]. However, the significance of this finding in the context of AD is currently unclear.

## 5. Applications of Patient-Derived Tau Seeding Models

As mentioned above, patient-derived tau seeding models present with considerable advantages over their Tg and viral counterparts. These models have not only confirmed previous findings relating to tau propagation, but also informed on the specific tau fibrillar structures and histopathological features differentiating tauopathies. Importantly, their use in different rodent models will prove crucial in two aspects. They can be used to elucidate disease mechanisms involved in the pathogenesis of AD and primary tauopathies by better recapitulating the disease-specific tau conformation and affected cell-types. This improved translational value may in turn pave the way for the discovery of successful tau-targeting therapeutics. Moreover, a deeper understanding of the different fibrillar tau species and their properties in the different tauopathies could be key to their use as diagnostics and the characterization thereof in these animal models.

### 5.1. Identification of Seed-Competent Tau Species

A large body of research has aimed at elucidating the tau species responsible for pathological tau spread in vivo. Patient-derived tau seeding models could be a useful tool to uncover seed-competent tau species, which will hopefully enable tau-targeted tauopathy treatments to be specifically tailored to the deleterious form of tau proteins in tauopathies. Takeda et al. (2015) [51] compared the propagation and uptake properties exhibited by a range of tau species derived from the brains of both transgenic tauopathy mice and human AD patients. These experiments consistently showed that soluble phosphorylated high molecular weight tau were the species involved in tau uptake and propagation [51]. However, oligomeric and sarkosyl-insoluble aggregated tau fractions isolated from human AD brains were also identified as seeds capable of inducing aggregation and pathological spread [59]. The precise nature of the tau species relevant for seeding and propagation is therefore still under investigation and may differ among individual AD patients [140].

### 5.2. Intracellular Mechanisms and Tau Aggregate Degradation

The mounting use of patient-derived tau seeding models has also provided insight into some other possible contributing factors to disease progression in AD and tauopathy patients. For instance, granulovacuolar degeneration bodies (GVBs)—a poorly understood vacuolar structure often found to accumulate in the brain of tauopathy patients—were formed in neurons in both WT and P301L Tg mice following injection of recombinant P301L tau and human brain extracts from AD, PSP, and PART patients [141]. Further, GVB load was suggested to correlate with pathological tau load, pointing at a potential intracellular consequence of tau seeding in neurons [141].

Recently, several lines of research have investigated tau disaggregation mechanisms as possibly playing a causal role in tau pathology progression. Darwich et al. (2021) [142] identified a mutation in the gene encoding valosin-containing protein (VCP) associated with autosomal dominant FTD. Moreover, they uncovered the role of VCP to disaggregate tau seeds in mice injected with human AD-tau. Mice bearing a mutation in the VCP-encoding gene exhibited elevated tau pathology upon inoculation [142]. Therefore, using patient-derived tau seeding models could prove invaluable in supporting these findings. Taken together, seeding models have shed light on some of the intracellular consequences of tau fibril accumulation, and dysfunction of disaggregation mechanisms.

### 5.3. Role of Astrocytes in Tau Spreading

The mechanisms involved in the spreading of glial tau pathology are also understudied. Narasimhan et al. (2019) [143] investigated glial involvement in human tau pathology spread in a neuronal tau knockdown model injected with human CBD and PSP brain extracts. In the absence of neuronal tau pathology, oligodendrocytes, but not astrocytes, exhibited the ability to propagate tau species. These findings point at the pivotal glial contribution—oligodendrocytes in particular—regarding pathological tau spread. These results indicate that developing therapies capable of targeting both neuronal and glial mechanisms could be effective against those diseases associated with glial tau pathology. Astrocytes have also been suggested to regulate tau pathology. Astrocytic transcription factor EB also plays a role in clearing the extracellular space of pathological tau species in Tg mice following seeding with PFFs [144], identifying also a potential key role for astrocytes in tau propagation. These findings illustrate that it may be worth further exploring and elucidating the involvement of glial cells in tauopathies (reviewed in [145]).

### 5.4. Role of Microglia in Tau Spreading

A range of in vivo evidence has pointed to the association of tau pathology with reactive immune cells in the brain, as well as elevated pro-inflammatory molecular burden as an indicator of neuroinflammation [146]. The most widely investigated neuroinflammatory component in AD remain microglia, the principal immune cells of the CNS, which have been widely investigated in the context of Aβ pathology. The innate immune system was linked to AD through genome-wide association studies, which revealed several implicated genetic variants [147,148]. In the developing brain, microglia are known to play a key role in synaptic pruning, a process likely re-activated in the diseased state, such that microglia act to phagocytose neurons and synapses bearing pathological tau aggregates [149,150]. It is thought that in disease, microglia lose the ability to maintain homeostasis and fail to clear unwanted protein pathologies, thereby enabling their accumulation and propagation across the brain (reviewed in [151]).

Asai et al. (2015) [97] provided the first evidence that microglia were tightly correlated with tau pathology with the observation that depletion of microglia halted tau propagation in an AAV-based tauopathy model. Additionally, primary microglia isolated from the brain of tauopathy patients were revealed to contain tau seeds [152]. This suggests a dependency of pathological tau propagation on microglia [97,153]. This phenomenon was supported by the discovery of microglial receptor *TREM2* as a genetic risk variant for AD [147,154]. Both TREM2 deficiency and mutation resulted in elevated tau seeding and spreading potencies, particularly to dystrophic neurites surrounding Aβ plaques [114]. The neuroinflammation-associated transcription factor NF-κB has further been identified as a driver of microglia-mediated tau spreading upon intracerebral inoculation of tau seeds in PS19 mice [155].

Collectively, these findings suggest that therapeutically targeting microglial defects could be a means to reduce pathological tau load in AD and in primary tauopathies. Seeding models present as a useful tool to investigate such mechanistic links.

### 5.5. Tau Pathology in Lewy Body Dementia

Neuronal tau uptake can occur through receptor-mediated endocytosis, for instance via heparan sulfate proteoglycans in conjunction with specific receptors like LRP1 [68,73]. Loss of function of the endocytosis regulator leucine-rich repeat kinase 2 (*LRRK2*) is known to impede tau internalization by neuronal cells [70]. A recent study identified LRRK2-regulated receptor-mediated endocytosis as the main uptake mechanism for monomeric and aggregated tau species upon application of whole-genome clustered regularly interspaced short palindromic repeats (CRISPR) knockout screens in human induced pluripotent stem cell (iPSC)-derived excitatory neurons [70]. Further, inhibition of LRRK2 kinase activity prevented tau uptake, confirming its pivotal role in tau internalization, but not tau pathology induction [156]. In Tg mice overexpressing the *LRRK2* mutation, pathological tau spread across neuroanatomically connected regions following intracerebral inoculation of AD brain extracts, adopting a preferential spreading towards anterograde brain regions, beyond those observed in WT mice [157]. Intriguingly, kinase-activating mutations in the *LRRK2* gene are the prevailing cause of familial Parkinson’s disease (PD) and the most common risk factor for its sporadic form, and carriers have been revealed to exhibit tau pathology [158]. Moreover, PD dementia (PDD) and dementia with Lewy bodies (DLB) have also been associated with the presence of intraneuronal tau aggregation, as in tauopathies, suggesting a role for LRRK2 in mediating tau spread [159,160].

PD, PDD, and DLB are classified as synucleinopathies, as they are all characterized by Lewy bodies mainly composed of aggregated α-synuclein (α-syn). Tau pathology observed in synucleinopathies is thought to correlate with cognitive decline and α-syn pathology [161]. Additionally, α-syn pathology has also been described in the brains of AD and PSP patients [162,163]. On this basis, Bassil et al., (2020) [164] proceeded to inject mouse α-syn PFFs, human AD-tau, or both into the brains of WT mice to elucidate the interaction between both pathologies. Then, α-syn PFFs were injected in tau knock-out (KO) mice, and conversely, AD-tau brain extracts were injected in α-syn KO mice. These experiments revealed that α-syn modulates tau spread and therefore overall pathological tau load. Seeding tau Tg animals with different α-syn seeds showed that specific α-syn strains could be more effective at promoting tau pathology than others [165].

All in all, the pathological overlap described here between tauopathies and LBD reveal mechanistic links between neurodegenerative disorders, which could be tackled in a more integrative fashion.

### 5.6. Tau-Targeted AD Treatments

Given the clinical failure of Aβ-directed therapies as disease-modifying treatments of AD, recent focus has been shifted to tau-targeted approaches. Tau-targeted immunotherapy has been widely investigated to block pathological spread mediated by extracellular tau species with minimal impact on intracellular biological functions. However, one major limitation to the development of a clinically useful approach lies in the fact that the optimal tau epitope for therapeutic efficacy remains elusive. Therefore, numerous monoclonal antibodies (mABs) targeting different tau protein domains and exhibiting differential selectivity have been investigated [166,167,168]. Testing those mABs intended to treat AD using the seeding models with AD brain lysates will provide unprecedented support for the efficacy of the immunotherapeutic approach.

Some of these mABs specifically bind phosphorylated tau protein residues. Indeed, approximately 85 potential phosphorylation sites have been identified on the tau protein, and abnormal phosphorylation is a known indicator of tau pathology. As such, administration of a range of mABs, such as the mAB C10.2, successfully reduced tau seeding when injected in Tg AD tau seeding mouse models [166,169]. Similarly, Dai et al. (2015, 2018) [170,171] treated 3xTg mice with peripheral injections of 43D, a mAB targeting the N-terminal region of tau. This resulted in a reduction in the level of hyperphosphorylated tau and blockage of inoculated AD patient tau seeding [170,171]. Recently, Courade et al. (2018) [167] developed a screening tool to identify the mAB that would be most effective against human tau seeds and found mAB D targeting the tau mid-region to display the highest activity. This mAB was later shown to successfully halt the progression of tau pathology following injection of human AD brain extracts in Tg mice expressing the human tau P301L mutation [172]. Gibbons et al. (2020) [173] later identified two additional mABs capable of inhibiting tau pathology induced upon human AD material inoculation in an aggressive amyloid pathology model (5xFAD mice). Confirming the efficacy of these treatments in models with better translational value, such as humanized tau models injected with patient-derived tau fibrils, could be beneficial.

As an alternative approach to immunotherapy, DeVos et al. (2017) [174] identified antisense oligonucleotides (ASOs), which effectively reduced human tau mRNA and protein load, in Tg mice of the PS19 line. This tau-lowering therapeutic approach is currently undergoing clinical trials. Similar to the aforementioned anti-tau mABs, it may be of interest to investigate the effect of ASOs in tau seeding models in the absence of endogenous tau pathology, which seemed to prevent injected tau seeding and subsequent spreading in a recent study [122].

## 6. Conclusions and Future Directions

In conclusion, injection of mouse models with patient-derived tau fibrils provides the most translationally relevant means to model disease mechanisms, test tau-targeted therapeutics, and tau-based diagnostics. The characteristic morphological features of tau protein aggregates, the cell-type specificity of pathological inclusions, and the differential spreading patterns in distinct tauopathies could underlie the heterogeneity in clinical presentations among tauopathy patients. Moreover, uncovering seed-competent species is crucial to developing disease-modifying treatments aimed at halting or slowing tau propagation. Most in vivo seeding studies have focused on injection of sarkosyl-insoluble brain extracts, but other methods such as injection of purified soluble tau oligomers could be explored further [120,135]. In addition, the functional consequences of tau oligomers could also warrant more investigation [175].

Passive immunotherapy with monoclonal antibodies directed against the relevant tau species presents as an attractive candidate, which would be worth testing in seeding models for enhanced validity. Animal models with good translational value are key to developing effective treatments. However, a recent study has highlighted the limitation of intracerebral injections of supraphysiological concentrations of tau fibrils [176]. Furthermore, patient to patient variability in terms of tau aggregate conformations, tau PTMs, affected brain regions, and clinical symptoms [76,140,177] may be additional limitations to these models, as this makes it difficult to reproduce findings between different research groups. However, the observation that patient-derived CSF induces seeding in transgenic mice may lead to improved patient stratification and better characterization of tau pathology in individual patients in vivo [116].

One should therefore be mindful that there is no perfect animal model of AD, and that the choice of relevant research tools is dependent on the specific question being investigated. Nevertheless, the increasing use of patient-derived brain lysates and the recent emergence of humanized tauopathy mouse models may be decisive in the search for a treatment capable of slowing disease progression or reversing its detrimental effects. Additionally, patient-derived tau seeding in rat models could also be further explored, as their larger brain size may make them better research models for characterizing tau PET tracers compared to mice [178,179]. Of note, robust cognitive deficits have thus far not been reported in seeding models. Optimization of brain extraction protocols to obtain more widespread tau pathology and deeper characterization of potential cognitive deficits should therefore be one of the focuses of future tau-related disease modelling research.

## Figures and Tables

**Figure 1 ijms-22-06132-f001:**
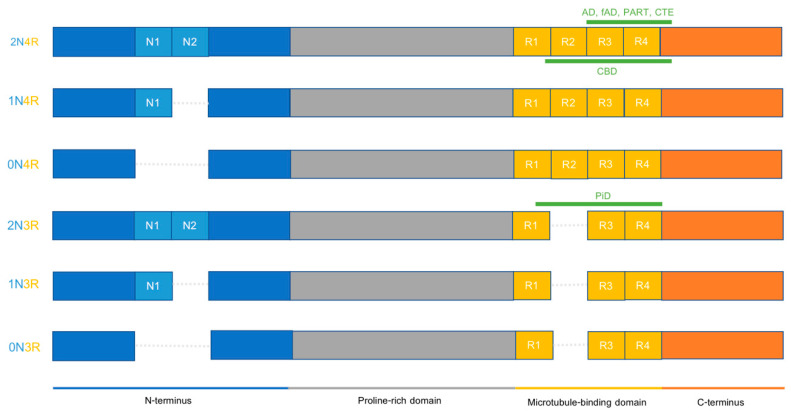
Tau isoforms in the adult human CNS and their relation to disease-specific tau fibrils. Tau filaments from distinct tauopathies exhibit different structural features. AD, fAD, and PART have identical cores spanning amino acids 306 to 378 that can incorporate all six tau isoforms (tau core was only indicated on the 2N4R isoform but applies to all six isoforms). These filament cores include repeats 3 and 4 and a portion of the C-terminus. The CTE core has some similarity to the AD core in terms of incorporated regions (R3 and R4) but is structurally distinct and is located between amino acids 305 to 379. CBD tau fibrils have a core that spans R1 until 12 residues after R4 and can only include 4R isoforms (tau core was only indicated on the 2N4R isoform but applies to all three 4R isoforms). CTE cores include amino acids 274 to 380. PiD tau fibrils have a core that spans R1 to R4 of 3R tau isoforms, between amino acids 254 and 378 (tau core was only indicated on the 2N3R isoform but applies to all three 3R isoforms). This core excludes R2, explaining why this is a 3R tauopathy.

**Table 1 ijms-22-06132-t001:** Summary table of the key studies involving injection of patient-derived tau fibrils in different mouse models.

		Tg Line	Tau Isoform	Tau Overexpression	Key Study
Wild-type mice	Mice expressing wild-type mouse tau		4R	No	Lasagna-Reeves et al., 2012 Guo et al., 2016 Ferrer et al., 2018, 2019, 2020, 2020, 2020 Narasimhan et al., 2017 Henderson et al., 2020
Tauopathy mouse models	Tg mice overexpressing human tau with FTD-linked mutation	PS19 (P301S MAPT mutation)	4R	Yes	Boluda et al., 2015
		T40PL-GFP (P301L MAPT mutation + GFP)	4R	Yes	Gibbons et al., 2017
	Tg mice overexpressing WT human tau	ALZ17	4R	Yes	Clavaguera et al., 2013
	Mice overexpressing humanized MAPT in the absence of mouse tau	hTau	3R > 4R	Yes	Hu et al., 2016 Miao et al., 2019
	Mice overexpressing all 6 human tau isoforms in the absence of mouse tau	6hTau	3R = 4R	Yes	He et al., 2020 Xu et al., 2021
Amyloid mouse models	Mice expressing wild-type mouse tau	5xFAD		No	He et al., 2018 Vergara et al., 2019 Xu et al., 2021
		APP KI		No	He et al., 2018
	Mice with targeted replacement of the mouse MAPT sequence	MAPT/APP double KI	3R > 4R	No	Saito et al., 2019

## Data Availability

Not applicable.

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
