# Peer review of "Tau Seeding Mouse Models with Patient Brain-Derived Aggregates"

_ijms, 2021, doi:10.3390/ijms22116132_

Round 1
Reviewer 1 Report
This manuscript submitted as a review by Robert, Schöll, and Vogels covers every aspect of Tau seeding reported in mouse models. The manuscript includes a lot of information that will be of interest to the International Journal of Molecular Sciences readership and the authors have done an excellent job at citing relevant references. Overall this is a very well-written manuscript and the publication is recommended after a minor revision. Authors are invited to address the following comments:
1) Page 2, Line 72: Cite at least one relevant article reporting the recent failures of Tau drugs in clinical trials.
2) Page 5, Figure 1: Why are there two green lines above CBD as opposed to one? Also, the authors may want to describe some more on figure legend.
3) Page 5, Line 188: abbreviation fAD occurs the first time here, so expand fAD here.
4) Page 7, Line 277: Replace “Thirdly” with “Fourthly”
Author Response
We thank the reviewers for their careful reading of our manuscript and for their thoughtful suggestions. We hope the reviewers agree that the manuscript has benefited substantially from their input. Below we respond to each comment on a point-by-point basis and changes are tracked in red in the manuscript (Please see the attachment for the revised version of the manuscript).
Reviewer 1:
“This manuscript submitted as a review by Robert, Schöll, and Vogels covers every aspect of Tau seeding reported in mouse models. The manuscript includes a lot of information that will be of interest to the International Journal of Molecular Sciences readership and the authors have done an excellent job at citing relevant references. Overall this is a very well-written manuscript and the publication is recommended after a minor revision.”
Reply: Thank you very much for your positive assessment of our manuscript.
“1) Page 2, Line 72: Cite at least one relevant article reporting the recent failures of Tau drugs in clinical trials.”
Reply: We have now cited a systematic review by Cummings et al., 2021, published last week, that describes the current AD pipeline (line 73, citation 22). We have also highlighted the failures of tau-targeting drugs in the associated sentence.
“2) Page 5, Figure 1: Why are there two green lines above CBD as opposed to one? Also, the authors may want to describe some more on figure legend.”
Reply: Thank you for this observation. We have now removed one of the green lines above CBD. We have also added more details to the figure legend (lines 219-229).
“3) Page 5, Line 188: abbreviation fAD occurs the first time here, so expand fAD here.”
Reply: We have added a sentence to the introduction to describe fAD (lines 30-32): Although most AD cases are sporadic, mutations in Aβ--related genes amyloid precursor protein (APP), presenilin (PSEN1), and PSEN2 lead to familial AD (fAD).
“4) Page 7, Line 277: Replace “Thirdly” with “Fourthly” “
Reply: Thank you. We have now replaced “Thirdly” with “Fourthly” (line 325)

Reviewer 2 Report
Summary
This review article provides an overview of the evolution of animal models of tau aggregation and propagation relevant to tauopathic neurodegenerative diseases; culminating in an argument for the importance of a translational model that best reflects human tauopathies. A model in which mouse tau is absent and all 6 human isoforms (or at least both 3R and 4R isoforms) are represented is desirable in order to provide the same monomeric tau building blocks that are available for oligomer and fibril progression in human disease. Seeding of tau aggregation in this model with brain-derived tau from tauopathy patient tissue appears to enable propagation of aggregate characteristics including structural features, cell-type specificity and patterns of spreading. Such models should be more translationally relevant for the evaluation tau-targeting therapeutics and tau-based diagnostics.
Broad Comments
This review provides a well-organized, insightful and relevant assessment of mouse models of tau aggregation; landing on the importance of a translational model that best recapitulates the human pathological condition in tauopathies. The review is well-written and clear. The upside of an AD brain derived tau-seeded human tau (3R/4R) animal model is clearly established.
The authors mention the heterogeneity of clinical presentation among tauopathy patients; clearly this is of benefit in that different tauopathies may be investigated and compared, as discussed. However, it may be useful to circle this back to some discussion regarding a potential advantages/disadvantages with model variability based on different human tissue samples within a single disease (like AD). Model variability in general may be disadvantageous. However, if CSF-derived tau seeding recapitulates tauopathic features, perhaps a personalized medicine approach would be eventually enabled that capitalizes on this.
The authors mention optimization of brain extraction protocols to improve the model severity to include cognitive defects. However, it seems that some discussion regarding the source/method/relevance of extraction of the human AD seeds could be warranted in the discussion as an aspect of the models themselves. It appears that the majority of brain derived tau seeds in your table are Sarkosyl-insoluble fractions, yet certain studies appear to use a different fraction (e.g. AD p-tau:oligomeric and hyperphosphorylated). Further, the use of immunopurified conformation-specific tau species derived from AD brain (e.g. T22 purified oligomers, Castillo-Carranza et al 2014) in an htau model may be relevant to the discussion of these models and/or future directions. In fact, any discussion of tau oligomers versus fibrils regarding which is the more toxic/pathogenic species or competency to seed aggregation in these models is notably absent.
Specific Comments
Lines 63-64: While these models have the potential to be translational, it is premature to describe them as overall better translational models if no successful therapeutics have been translated.
Section 2.1: References are mostly reviews; it would be useful to include some key original articles.
Line 100: Here the implications of phosphorylation and truncation are described as widely unknown. Certainly it is far from nailed down, especially with so many potential permutations of phosphorylation status due to many phosphorylation sites. But this comment appears to gloss over the demonstrated effects of phosphorylation and truncation to favor aggregation. It also appears to be inconsistent with discussion of phosphorylation in lines 682-684.
Section 3.1: Additional limitation or potential problems with Tg models could be include, like endogenous gene disruptions, drift of the model, sex differences, etc.
Lines 236-238: EC is an early affected brain region. locus coeruleus is the first known affected brain region.
Lines 246-247: This should be written to be explicit that the authors are referring to the Tg models discussed in this section only rather than “Tg models”.
Lines 163-171: In this discussion if 3R and 4R isoforms in tauopathies, the term FTD appears to be notably absent. Perhaps it would be useful to explicitly state somewhere which tauopathies are classified as FTD to be clear to a variety of readers.
Figure 1: There appear to be two green lines below 2N4R with only one label of “CBD”
Line 188: familial AD should be defined at first use.
Lines 257-8: better to state one human tau isoform….Which isoform? (0N4R?)
Line 263: “ the most important limitation lies in their practicality” regarding the time and expense of the studies? If the goal is the best translational model for a slowly progressing disease of aging, this is probably not something to be avoided but embraced. The most important limitations would be things that stray further from being a good translational model…
Lines 283-5: this statement follows the absence of endogenous mouse tau (KO) reference. Therefore it should be made clear that ref 80 is in the htau model.
Lines 287-88: Not all Tg tau mice models express the transgene from birth. Regulatable (tet) Tg models have also been evaluated.
Lines 296-7: Sounds future tense, but then results are discussed.
Lines 299-300: Similar to AD pathology in what way? Silver staining? aggregate structure?
Lines 301-312: Identifying the single isoform in this study as 2N4R would be appropriate;
What is in pathology-affected brain extract… hyperphosphorylated tau fibrils? In this study the brain extract is different than Sarkosyl Insoluble Fractions used in many other studies.
Lines 313-321: First sentence could be written more clearly to clarify that is P301S brain extracts, into the same P301S transgenic line. Also that the brain extracts were from aged brain (5.5 months) and injected into younger animals (2 months). This study also makes use of brain extracts for seeding rather than insoluble fractions.
Lines 329-336: One study with recombinant seeding is mentioned, but there is more literature in the recombinant seeding models that could be explored here.
Lines 347-8: Again translational potential may be better, but it is hard to say it is a translationally improved alternative when nothing has translated.
Lines 421-7: Was there any tau aggregation/pathology present in this study?
Line 433: AD-tau was found to not be present after 7 days, suggesting this is not a concern. Guo 2016
Lines 440-442: Should include a reference to make it clear that this statement refers to the same study, Guo 2016, as it sounds like you are introducing a new study.
Lines 466-472: Uchihara 2014 describes the importance of 3R in the maturation of tau pathology which may be relevant to this section.
Section 4.4 Castillo-Carranza et al 2014 should be included in this section
Lines 517-524: No references for this model are cited. Belfiore et al 2019
Lines 539-548: would be useful to mention isoform expression/ratios here for context
Line 684: inconsistent with Line 100
Lines 702-705: The second half of the sentence could be written more clearly or explicitly. It is unclear whether the reference matches the statement made here.
Lines 717-718: Here supraphysiological concentrations are mentioned as a potential limitation as a translational model. Patient to patient variability regarding the nature of the source of tau seeds as well as brain region heterogeneity may be additional limitation/variability of the model.
Table: The following references would make suitable additions to your table
Lasagna-Reeves et al 2012, Reference 96,
Xu et.al 2021 PMCID: PMC7847465
Vergara et al. 2019 PMID: 30599077
Author Response
We thank the reviewers for their careful reading of our manuscript and for their thoughtful suggestions. We hope the reviewers agree that the manuscript has benefited substantially from their input. Below we respond to each comment on a point-by-point basis and changes are tracked in red in the manuscript (Please see the attachment for the revised version of the manuscript).
Reviewer 2:
Broad Comments
“The authors mention the heterogeneity of clinical presentation among tauopathy patients; clearly this is of benefit in that different tauopathies may be investigated and compared, as discussed. However, it may be useful to circle this back to some discussion regarding a potential advantages/disadvantages with model variability based on different human tissue samples within a single disease (like AD). Model variability in general may be disadvantageous. However, if CSF-derived tau seeding recapitulates tauopathic features, perhaps a personalized medicine approach would be eventually enabled that capitalizes on this.”
Reply: Thank you for this interesting point. We have addressed this in the conclusion (lines 778-783): Furthermore, patient to patient variability in terms of tau aggregate conformations, tau PTMs, affected brain regions, and clinical symptoms (76,141,178) may be additional limitations to these models, as this makes it difficult to reproduce findings between different research groups. However, the observation that patient-derived CSF induces seeding in transgenic mice may lead to improved patient stratification and better characterisation of tau pathology in individual patients in vivo (117).
“The authors mention optimization of brain extraction protocols to improve the model severity to include cognitive defects. However, it seems that some discussion regarding the source/method/relevance of extraction of the human AD seeds could be warranted in the discussion as an aspect of the models themselves. It appears that the majority of brain derived tau seeds in your table are Sarkosyl-insoluble fractions, yet certain studies appear to use a different fraction (e.g. AD p-tau:oligomeric and hyperphosphorylated). Further, the use of immunopurified conformation-specific tau species derived from AD brain (e.g. T22 purified oligomers, Castillo-Carranza et al 2014) in an htau model may be relevant to the discussion of these models and/or future directions. In fact, any discussion of tau oligomers versus fibrils regarding which is the more toxic/pathogenic species or competency to seed aggregation in these models is notably absent.”
Reply: We have now addressed this in section 2.2 (line 137-141): It is still an outstanding question which forms of tau are the predominant seed-competent species in tauopathy patients. Misfolded monomers, soluble hyper-phosphorylated tau aggregates, and short fibrils have all been proposed to be responsible for tau seeding (29,51,52). Interestingly, differences in seed-competence of soluble hyperphosphorylated tau species was correlated to clinical heterogeneity in AD pa-tients (28,51,52).
In addition, we have added the following sentences to the conclusion (line 768-774): Most in vivo seeding studies have focused on injection of sarkosyl insoluble brain extracts, but other methods such as injection of purified soluble tau oligomers could be explored further (121,136). In addition, the functional consequences of tau oligomers could also warrant more investigation (176).
Specific Comments
“Lines 63-64: While these models have the potential to be translational, it is premature to describe them as overall better translational models if no successful therapeutics have been translated.”
Reply: We have now described them as follows: Finally, patient-derived seeding models could potentially be more translationally relevant models … (lines 65-66)
“Section 2.1: References are mostly reviews; it would be useful to include some key original articles.”
Reply: We have now added key original references to section 2.1 (highlighted in red).
“Line 100: Here the implications of phosphorylation and truncation are described as widely unknown. Certainly it is far from nailed down, especially with so many potential permutations of phosphorylation status due to many phosphorylation sites. But this comment appears to gloss over the demonstrated effects of phosphorylation and truncation to favor aggregation. It also appears to be inconsistent with discussion of phosphorylation in lines 682-684.”
Reply: We have added a more detailed section on tau PTMs (lines 101-123).
“Section 3.1: Additional limitation or potential problems with Tg models could be include, like endogenous gene disruptions, drift of the model, sex differences, etc.”
Reply: The following points have been added (line 265-270): Importantly, transgenic models are associated with phenotypic artefacts not related to tau pathology, caused by the disruption of endogenous genes and genetic drift. (95). The genome and phenotype of both existing and new transgenic models therefore needs to be carefully characterised. Furthermore, sex differences have been observed in trans-genic tauopathy mic (96). However, these are often not explicitly reported or investigated, which could be an important focus of future research.
“Lines 236-238: EC is an early affected brain region. locus coeruleus is the first known affected brain region.”
Reply: We have addressed this as follows (line 272-274) : Tg models with localised tau pathology were designed to selectively express mu-tated human tau in the EC, which along with the locus coeruleus is one of the earliest brain regions affected by tau pathology in AD
“Lines 246-247: This should be written to be explicit that the authors are referring to the Tg models discussed in this section only rather than “Tg models”.”
Reply: We have now clarified this in the manuscript (section 3.2, highlighted in red)
“Lines 163-171: In this discussion if 3R and 4R isoforms in tauopathies, the term FTD appears to be notably absent. Perhaps it would be useful to explicitly state somewhere which tauopathies are classified as FTD to be clear to a variety of readers.”
Reply: This has now been clarified in the introduction section (lines 34-38)
“Figure 1: There appear to be two green lines below 2N4R with only one label of “CBD””
Reply: Thank you, this has been corrected.
“Line 188: familial AD should be defined at first use.”
Reply: We have added a sentence to the introduction to describe fAD (lines 30-32): Although most AD cases are sporadic, mutations in Aβ--related genes amyloid precursor protein (APP), presenilin (PSEN1), and PSEN2 lead to familial AD (fAD).
“Lines 257-8: better to state one human tau isoform….Which isoform? (0N4R?)”
Reply: This has now been added (line 295-296)
“Line 263: “ the most important limitation lies in their practicality” regarding the time and expense of the studies? If the goal is the best translational model for a slowly progressing disease of aging, this is probably not something to be avoided but embraced. The most important limitations would be things that stray further from being a good translational model…”
Reply: We have decided to remove this sentence.
“Lines 283-5: this statement follows the absence of endogenous mouse tau (KO) reference. Therefore it should be made clear that ref 80 is in the htau model.”
Reply: This has now been clarified (lines 316-321)
“Lines 287-88: Not all Tg tau mice models express the transgene from birth. Regulatable (tet) Tg models have also been evaluated.”
Reply: This has now been clarified (lines 322-324)
“Lines 296-7: Sounds future tense, but then results are discussed.”
Reply: This has now been corrected (lines 332).
“Lines 299-300: Similar to AD pathology in what way? Silver staining? aggregate structure?”
Reply: This has been clarified as follows (lines 335-336) : ). Whether this method induces tau filaments with a structure similar to AD remains to be determined
“Lines 301-312: Identifying the single isoform in this study as 2N4R would be appropriate; What is in pathology-affected brain extract… hyperphosphorylated tau fibrils? In this study the brain extract is different than Sarkosyl Insoluble Fractions used in many other studies.”
Reply: This has been clarified as follows (lines 340-343) : They injected tau filament-containing brain homogenate from a Tg mouse model that expresses the 0N4R human tau isoform with the FTD-linked P301S mutation into the hippocampus and overlying cerebral cortex of Tg mice overexpressing a single WT human tau isoform (2N4R).
“Lines 313-321: First sentence could be written more clearly to clarify that is P301S brain extracts, into the same P301S transgenic line. Also that the brain extracts were from aged brain (5.5 months) and injected into younger animals (2 months). This study also makes use of brain extracts for seeding rather than insoluble fractions.”
Reply: The following changes were made (lines 350-352): Similarly, Ahmed et al., 2014 (107) proceeded to inject brain extracts of 5.5 month old P301S transgenic mice for seeding into the brain of mice from the same line at 2 months of age
“Lines 329-336: One study with recombinant seeding is mentioned, but there is more literature in the recombinant seeding models that could be explored here.”
Reply: We have added a section on recombinant tau seeding models (375-378)
“Lines 347-8: Again translational potential may be better, but it is hard to say it is a translationally improved alternative when nothing has translated.”
Reply: This has been amended: These models have greater translational potential compared to… (lines 389-390)
“Lines 421-7: Was there any tau aggregation/pathology present in this study?”
Reply: The presence of AT8-positive inclusions in neurons was addressed as follows (lines 365): …exhibited inclusions containing hyperphosphorylated tau…
“Line 433: AD-tau was found to not be present after 7 days, suggesting this is not a concern. Guo 2016”
Reply: We have decided to remove this sentence.
“Lines 440-442: Should include a reference to make it clear that this statement refers to the same study, Guo 2016, as it sounds like you are introducing a new study.”
Reply: The citations have been added these lines have been clarified (lines 478-484)
“Lines 466-472: Uchihara 2014 describes the importance of 3R in the maturation of tau pathology which may be relevant to this section.”
Reply: We have added this sentence (lines 511-512): Of note, the presence of 3R tau isoforms has been suggested as a key element in the maturation of tau pathology (130).
“Section 4.4 Castillo-Carranza et al 2014 should be included in this section”
Reply: We have added this sentence and reference (lines 554-555): Moreover, they present as a model with great translational potential for testing tau-targeted drug treatments (See Section 5.6), (136).
“Lines 517-524: No references for this model are cited. Belfiore et al 2019”
Reply: We have added the following citations (Line 566): Oddo et al., 2003, Belfiore et al., 2019 (137,138)
“Lines 539-548: would be useful to mention isoform expression/ratios here for context"
Reply: We have added the following sentence (lines 585-587): These mice are characterized by A pathology and express all 6 isoforms of human tau – with 3R tau isoforms slightly more abundant compared to 4R isoforms.
“Line 684: inconsistent with Line 100”
Reply: We have added additional paragraphs on tau PTMs to make these sections consistent (lines 101-123).
“Lines 702-705: The second half of the sentence could be written more clearly or explicitly. It is unclear whether the reference matches the statement made here.”
Reply: This has now been clarified (lines 745-751)
“Lines 717-718: Here supraphysiological concentrations are mentioned as a potential limitation as a translational model. Patient to patient variability regarding the nature of the source of tau seeds as well as brain region heterogeneity may be additional limitation/variability of the model.”
Reply: This has now been added (lines 768-774): Furthermore, patient to patient variability in terms of tau aggregate conformations, tau PTMs, affected brain regions, and clinical symptoms (76,141,178) may be additional limitations to these models, as this makes it difficult to reproduce findings between different research groups.
“Table: The following references would make suitable additions to your table
Lasagna-Reeves et al 2012, Reference 96,
Xu et.al 2021 PMCID: PMC7847465
Vergara et al. 2019 PMID: 30599077”
Reply: These references have now been added to the table.
